# Long-Tailed Classification by Keeping the Good and Removing the Bad Momentum Causal Effect

**Kaihua Tang[1], Jianqiang Huang[1,2], Hanwang Zhang[1]**
[1]Nanyang Technological University, [2]Damo Academy, Alibaba Group
kaihua001@e.ntu.edu.sg, jianqiang.jqh@gmail.com, hanwangzhang@ntu.edu.sg

## Abstract

As the class size grows, maintaining a balanced dataset across many classes is challenging because the data are long-tailed in nature; it is even impossible when the sample-of-interest co-exists with each other in one collectable unit, *e.g.*, multiple visual instances in one image. Therefore, long-tailed classification is the key to deep learning at scale. However, existing methods are mainly based on re-weighting/re-sampling heuristics that lack a fundamental theory. In this paper, we establish a causal inference framework, which not only unravels the whys of previous methods, but also derives a new principled solution. Specifically, our theory shows that the SGD momentum is essentially a confounder in long-tailed classification. On one hand, it has a harmful causal effect that misleads the tail prediction biased towards the head. On the other hand, its induced mediation also benefits the representation learning and head prediction. Our framework elegantly disentangles the paradoxical effects of the momentum, by pursuing the direct causal effect caused by an input sample. In particular, we use causal intervention in training, and counterfactual reasoning in inference, to remove the "bad" while keep the "good". We achieve new state-of-the-arts on three long-tailed visual recognition benchmarks[1]: Long-tailed CIFAR-10/-100, ImageNet-LT for image classification and LVIS for instance segmentation.

## 1   Introduction

Over the years, we have witnessed the fast development of computer vision techniques [1, 2, 3], stemming from large and balanced datasets such as ImageNet [4] and MS-COCO [5]. Along with the growth of the digital data created by us, the crux of making a large-scale dataset is no longer about where to collect, but how to balance. However, the cost of expanding them to a larger class vocabulary with balanced data is not linear — but exponential — as the data will be inevitably long-tailed by Zipf's law [6]. Specifically, a single sample increased for one data-poor tail class will result in more samples from the data-rich head. Sometimes, even worse, re-balancing the class is impossible. For example, in instance segmentation [7], if we target at increasing the images of tail class instances like "remote controller", we have to bring in more head instances like "sofa" and "TV" simultaneously in every newly added image [8].

Therefore, long-tailed classification is indispensable for training deep models at scale. Recent work [9, 10, 11] starts to fill in the performance gap between class-balanced and long-tailed datasets, while new long-tailed benchmarks are springing up such as Long-tailed CIFAR-10/-100 [12, 10], ImageNet-LT [9] for image classification and LVIS [7] for object detection and instance segmentation. Despite the vigorous development of this field, we find that the fundamental theory is still missing. We conjecture that it is mainly due to the paradoxical effects of long tail. On one hand, it is bad because

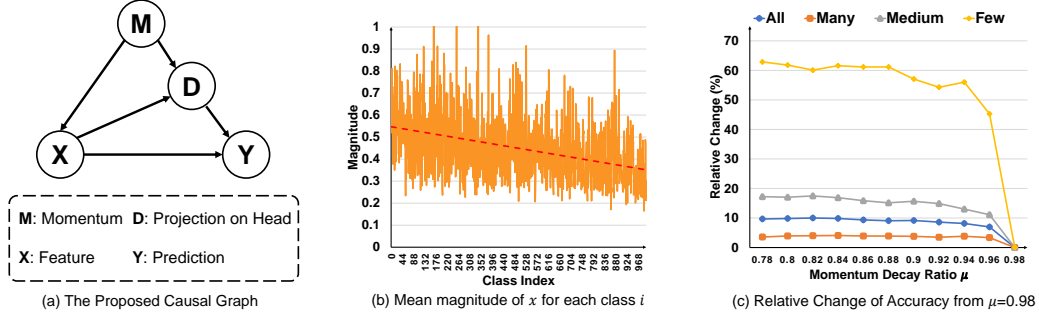

(a) The Proposed Causal Graph

(b) Mean magnitude of $x$ for each class $i$

(c) Relative Change of Accuracy from $\mu=0.98$

Figure 1: (a) The proposed causal graph explaining the causal effect of momentum. See Section 3 for details. (b) The mean magnitudes of feature vectors for each class $i$ after training with momentum $\mu = 0.9$, where $i$ is ranking from head to tail. (c) The relative change of the performance on the basis of $\mu = 0.98$ shows that the few-shot tail is more vulnerable to the momentum.

the classification is severely biased towards the data-rich head. On the other hand, it is good because the long-tailed distribution essentially encodes the natural inter-dependencies of classes — "TV" is indeed a good context for "controller" — any disrespect of it will hurt the feature representation learning [10], *e.g.*, re-weighting [13, 14] or re-sampling [15, 16] inevitably causes under-fitting to the head or over-fitting to the tail.

Inspired by the above paradox, latest studies [10, 11] show promising results in disentangling the "good" from the "bad", by the naïve two-stage separation of *imbalanced* feature learning and *balanced* classifier training. However, such disentanglement does not explain the whys and wherefores of the paradox, leaving critical questions unanswered: given that the re-balancing causes under-fitting/over-fitting, why is the re-balanced classifier good but the re-balanced feature learning bad? The two-stage design clearly defies the end-to-end merit that we used to believe since the deep learning era; but why does the two-stage training significantly outperform the end-to-end one in long-tailed classification?

In this paper, we propose a causal framework that not only fundamentally explains the previous methods [15, 16, 17, 9, 11, 10], but also provides a principled solution to further improve long-tailed classification. The proposed causal graph of this framework is given in Figure 1 (a). We find that the momentum $M$ in any SGD optimizer [18, 19] (also called betas in Adam optimizer [20]), which is indispensable for stabilizing gradients, is a confounder who is the common cause of the sample feature $X$ (via $M \rightarrow X$) and the classification logits $Y$ (via $M \rightarrow D \rightarrow Y$). In particular, $D$ denotes the $X$'s projection on the head feature direction that eventually deviates $X$. We will justify the graph later in Section 3. Here, Figure 1 (b&c) sheds some light on how the momentum affects the feature $X$ and the prediction $Y$. From the causal graph, we may revisit the "bad" long-tailed bias in a causal view: the backdoor [21] path $X \leftarrow M \rightarrow D \rightarrow Y$ causes the spurious correlation even if $X$ has nothing to do with the predicted $Y$, *e.g.*, misclassifying a tail sample to the head. Also, the mediation [22] path $X \rightarrow D \rightarrow Y$ mixes up the pure contribution made by $X \rightarrow Y$. For the "good" bias, $X \rightarrow D \rightarrow Y$ respects the inter-relationships of the semantic concepts in classification, that is, the head class knowledge contributes a reliable evidence to filter out wrong predictions. For example, if a rare sample is closer to the head class "TV" and "sofa", it is more likely to be a living room object (*e.g.*, "remote controller") but not an outdoor one (*e.g.*, "car").

Based on the graph that explains the paradox of the "bad" and "good", we propose a principled solution for long-tailed classification. It is a natural derivation of pursuing the direct causal effect along $X \rightarrow Y$ by removing the momentum effect. Thanks to causal inference [23], we can elegantly keep the "good" while remove the "bad". First, to learn the model parameters, we apply de-confounded training with causal intervention: while it removes the "bad" by *backdoor adjustment* [21] who cuts off the backdoor confounding path $X \leftarrow M \rightarrow D \rightarrow Y$, it keeps the "good" by retaining the mediation $X \rightarrow D \rightarrow Y$. Second, we calculate the direct causal effect of $X \rightarrow Y$ as the final prediction logits. It disentangles the "good" from the "bad" in a *counterfactual* world, where the bad effect is considered as the $Y$'s indirect effect when $X$ is zero but $D$ retains the value when $X = \boldsymbol{x}$. In contrast to the prevailing two-stage design [11] that requires unbiased re-training in the 2nd stage, our solution is one-stage and re-training free. Interestingly, as discussed in Section 4.4, we show that why the re-training is inevitable in their method and why ours can avoid it with even better performance.

On image classification benchmarks Long-tailed CIFAR-10/-100 [12, 10] and ImageNet-LT [9], we outperform previous state-of-the-arts [10, 11] on all splits and settings, showing that the performance gain is not merely from catering to the long tail or a specific imbalanced distribution. In object detection and instance segmentation benchmark LVIS [7], our method also has a significant advantage over the former winner [17] of LVIS 2019 challenge. We achieve 3.5% and 3.1% absolute improvements on mask AP and box AP using the same Cascade Mask R-CNN with R101-FPN backbone [24].

## 2   Related Work

**Re-Balanced Training.** The most widely-used solution for long-tailed classification is arguably to re-balance the contribution of each class in the training phase. It can be either achieved by re-sampling [25, 26, 15, 16, 27] or re-weighting [13, 14, 12, 17]. However, they inevitably cause the under-fitting/over-fitting problem to head/tail classes. Besides, relying on the accessibility of data distribution also limits their application scope, *e.g.*, not applicable in online and streaming data.

**Hard Example Mining.** The instance-level re-weighting [28, 29, 30] is also a practical solution. Instead of hacking the prior distribution of classes, focusing on the hard samples also alleviates the long-tailed issue, *e.g.*, using meta-learning to find the conditional weights for each samples [31], enhancing the samples of hard categories by group softmax [32].

**Transfer Learning/Two-Stage Approach.** Recent work shows a new trend of addressing the long-tailed problem by transferring the knowledge from head to tail. The sharing bilateral-branch network [10], the two-stage training [11], the dynamic curriculum learning [33] and the transferring memory features [9] / head distributions [34] are all shown to be effective in long-tailed recognition, yet, they either significantly increase the parameters or require a complicated training strategy.

**Causal Inference.** Causal inference [23, 35] has been widely adopted in psychology, politics and epidemiology for years [36, 37, 38]. It doesn't just serve as an interpretation framework, but also provides solutions to achieve the desired objectives by pursing causal effect. Recently, causal inference has also attracted increasing attention in computer vision society [39, 40, 41, 42, 43, 44] for removing the dataset bias in domain-specific applications, *e.g.*, using pure direct effect to capture the spurious bias in VQA [41] and NWGM for Captioning [42]. Compared to them, our method offers a fundamental framework for general long-tailed visual recognition.

## 3   A Causal View on Momentum Effect

To systematically study the long-tailed classification and how momentum affects the prediction, we construct a **causal graph** [23, 22] in Figure 1 (a) with four variables: momentum ($M$), object feature ($X$), projection on head direction ($D$), and model prediction ($Y$). The causal graph is a directed acyclic graph used to indicate how variables of interest $\{M, X, D, Y\}$ interacting with each other through causal links. The nodes $M$ and $D$ constitute a confounder and a mediator, respectively. A *confounder* is a variable that influences both correlated and independent variables, creating a spurious statistical correlation. Considering a causal graph **exercise** $\leftarrow$ **age** $\rightarrow$ **cancer**, the elder people spend more time on physical exercise after retirement and they are also easier to get cancer due to the elder age, so the confounder $age$ creates a spurious correlation that more physical exercise will increase the chance of getting cancer. The example of a *mediator* would be **drug** $\rightarrow$ **placebo** $\rightarrow$ **cure**, where mediator $placebo$ is the side effect of taking $drug$ that prevents us from getting the direct effect of **drug** $\rightarrow$ **cure**.

Before we delve into the rationale of our causal graph, let's take a brief review on the SGD with momentum [19]. Without loss of generality, we adopt the Pytorch implementation [45]:

$$v_t = \underbrace{\mu \cdot v_{t-1}}_{momentum} + g_t, \quad \theta_t = \theta_{t-1} - lr \cdot v_t, \tag{1}$$

where the notations in the $t$-th iteration are: model parameters $\theta_t$, gradient $g_t$, velocity $v_t$, momentum decay ratio $\mu$, and learning rate $lr$. Other versions of SGD [18, 19] only change the position of some hyper-parameters and we can easily prove them equivalent with each other. The use of momentum considerably dampens the oscillations caused by each single sample. In our causal graph, momentum $M$ is the overall effect of $\mu \cdot v_{T-1}$ at the convergence $t = T$, which is the exponential moving average of the gradient over all past samples with decay rate $\mu$. Eq. (1) shows that, given fixed

hyper-parameters $\mu$ and $lr$, each sample $M = \boldsymbol{m}$ is a function of the model initialization and the mini-batch sampling strategy, that is, $M$ has infinite samples.

In a balanced dataset, the momentum is equally contributed by every class. However, when the dataset is long-tailed, it will be dominated by the head samples, emerging the following causal links:

$M \to X$. This link says that the backbone parameters used to generate feature vectors $X$, are trained under the effect of $M$. This is obvious from Eq. (1) and can be illustrated in Figure 1 (b), where we visualize how the magnitudes of $X$ change from head to tail.

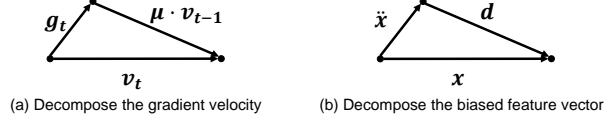

(a) Decompose the gradient velocity     (b) Decompose the biased feature vector

Figure 2: Based on Assumption 1, the feature vector $\boldsymbol{x}$ can be decomposed into a discriminative feature $\ddot{\boldsymbol{x}}$ and a projection on head direction $\boldsymbol{d}$

$(M, X) \to D$. This link denotes that the momentum also causes feature vector $X$ deviates to the head direction $D$, which is also determined by $M$. In a long-tailed dataset, few head classes possess most of the training samples, who have less variance than the data-poor but class-rich tail, so the moving averaged momentum will thus point to a stable head direction. Specifically, as shown in Figure 2, we can decompose any feature vector $\boldsymbol{x}$ into $\boldsymbol{x} = \ddot{\boldsymbol{x}} + \boldsymbol{d}$, where $D = \boldsymbol{d} = \hat{\boldsymbol{d}} cos(\boldsymbol{x}, \hat{\boldsymbol{d}}) \|\boldsymbol{x}\|$. In particular, the head direction $\hat{\boldsymbol{d}}$ is given in Assumption 1, whose validity is detailed in Appendix A.

**Assumption 1** *The head direction $\hat{\boldsymbol{d}}$ is the unit vector of the exponential moving average features with decay rate $\mu$ like momentum, i.e., $\hat{\boldsymbol{d}} = \overline{\boldsymbol{x}}_T / \|\overline{\boldsymbol{x}}_T\|$, where $\overline{\boldsymbol{x}}_t = \mu \cdot \overline{\boldsymbol{x}}_{t-1} + \boldsymbol{x}_t$ and $T$ is the number of the total training iterations.*

Note that Assumption 1 says that the head direction is exactly determined by the sample moving average in the dataset, which does not need the accessibility of the class statistics at all. In particular, as we show in Appendix A, when the dataset is balanced, Assumption 1 also holds but suggests that $X \to Y$ is naturally not affected by $M$.

$X \to D \to Y$ & $X \to Y$. These links indicate that the effect of $X$ can be disentangled into an indirect (mediation) and a direct effect. Thanks to the above orthogonal decomposition: $\boldsymbol{x} = \ddot{\boldsymbol{x}} + \boldsymbol{d}$, the indirect effect is affected by $\boldsymbol{d}$ while the direct effect is affected by $\ddot{\boldsymbol{x}}$, and they together determine the total effect. As shown in Figure 4, when we change the scale parameter $\alpha$ of $\boldsymbol{d}$, the performance of the tail classes monotonically increases with $\alpha$, which inspires us to remove the mediation effect of $D$ in Section 4.2.

# 4  The Proposed Solution

Based on the proposed causal graph in Figure 1 (a), we can delineate our goal for long-tailed classification: the pursuit of the direct causal effect along $X \to Y$. In causal inference, it is defined as Total Direct Effect (TDE) [46, 22]:

$$\arg\max_{i \in C} \ TDE(Y_i) = [Y_{\boldsymbol{d}} = i | do(X = \boldsymbol{x})] - [Y_{\boldsymbol{d}} = i | do(X = \boldsymbol{x}_0)], \qquad (2)$$

where $\boldsymbol{x}_0$ denotes a null input (0 in this paper). We define the causal effect as the prediction logits $Y_i$ for the $i$-th class. Subscript $\boldsymbol{d}$ denotes that the mediator $D$ always takes the value $\boldsymbol{d}$ in the *deconfounded* causal graph model of Figure 1 (a) with $do(X = \boldsymbol{x})$, where the $do$-operator denotes the causal intervention [23] that modifies the graph by $M \not\to X$. Thus, Eq. (2) shows an important principle in long-tailed classification: before we calculate the final TDE (Section 4.2), we need to first perform de-confounded training (Section 4.1) to estimate the "modified" causal graph parameters.

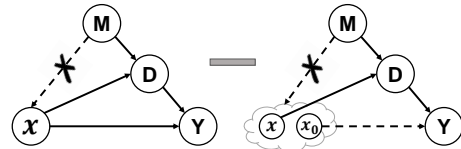

Figure 3: The TDE inference (Eq. (2)) for the long-tailed classification after de-confounded training. Subtracted left: $[Y_{\boldsymbol{d}} = i | do(X = \boldsymbol{x})]$, minus right: $[Y_{\boldsymbol{d}} = i | do(X = \boldsymbol{x}_0)]$.

We'd like to highlight that Eq. (2) removes the "bad" while keeps the "good" in a reconcilable way. First, in

training, the $do$-operator removes the "bad" confounder bias while keeps the "good" mediator bias, because the $do$-operator retains the mediation path. Second, in inference, the mediator value $\boldsymbol{d}$ is imposed in both terms to keep the "good" of the mediator bias (towards head) in logit prediction; it also removes its "bad" by subtracting the second term: the prediction when the input $X$ is null $(\boldsymbol{x}_0)$ but the mediator $D$ is still the value $\boldsymbol{d}$ when $X$ had been $\boldsymbol{x}$. Note that such a *counterfactual* minus elegantly characterizes the "bad" mediation bias, just like how we capture the tricky placebo effect: we cheat the patient to take a placebo drug, setting the direct drug effect **drug** $\rightarrow$ **cure** to zero; thus, any cure observed must be purely due to the non-zero placebo effect **drug** $\rightarrow$ **placebo** $\rightarrow$ **cure**.

## 4.1 De-confounded Training

The model for the proposed causal graph is optimized under the causal intervention $do(X = \boldsymbol{x})$, which aims to preserve the "good" feature learning from the momentum and cut off its "bad" confounding effect. We apply the backdoor adjustment [21] to derive the de-confounded model:

$$P(Y = i|do(X = \boldsymbol{x})) = \sum_{\boldsymbol{m}} P(Y = i|X = \boldsymbol{x}, M = \boldsymbol{m})P(M = \boldsymbol{m}) \tag{3}$$

$$= \sum_{\boldsymbol{m}} \frac{P(Y = i, X = \boldsymbol{x}|M = \boldsymbol{m})P(M = \boldsymbol{m})}{P(X = \boldsymbol{x}|M = \boldsymbol{m})}. \tag{4}$$

As there are infinite number of $M = \boldsymbol{m}$, it is prohibitively to achieve the above backdoor adjustment. Fortunately, the Inverse Probability Weighting [23] formulation in Eq. (4) provides us a new perspective in approximating the infinite sampling $(i, \boldsymbol{x}, \boldsymbol{d})|\boldsymbol{m}$. For a finite dataset, no matter how many $\boldsymbol{m}$ there are, we can only observe one $(i, \boldsymbol{x}, \boldsymbol{d})$ given one $\boldsymbol{m}$. In such cases, the number of $\boldsymbol{m}$ values that Eq. (4) would encounter is equal to the number of samples $(i, \boldsymbol{x}, \boldsymbol{d})$ available, not to the number of possible $\boldsymbol{m}$ values, which is prohibitive. In fact, thanks to the backdoor adjustment, which connects the equivalence between the originally confounded model $P$ and the deconfounded model $P$ with $do(X)$, we can collect samples from the former, that act as though they were drawn from the latter. Therefore, Eq. (4) can be approximated as

$$P(Y = i|do(X = \boldsymbol{x})) \approx \frac{1}{K} \sum_{k=1}^{K} \widetilde{P}(Y = i, X = \boldsymbol{x}^k, D = \boldsymbol{d}^k), \tag{5}$$

where $\widetilde{P}$ is the inverse weighted probability and we drop $M = \boldsymbol{m}$ for notation simplicity and bear in mind that $\boldsymbol{d}$ still depends on $\boldsymbol{m}$. In particular, compared to the vanilla trick, we apply a multi-head strategy [47] to equally divide the channel (or dimensions) of weights and features into $K$ groups, which can be considered as $K$ times more fine-grained sampling.

We model $\widetilde{P}$ in Eq. (5) as the softmax activated probability of the energy-based model [48]:

$$\widetilde{P}(Y = i, X = \boldsymbol{x}^k, D = \boldsymbol{d}^k) \propto E(\boldsymbol{x}^k, \boldsymbol{d}^k; \boldsymbol{w}_i^k) = \tau \frac{f(\boldsymbol{x}^k, \boldsymbol{d}^k; \boldsymbol{w}_i^k)}{g(\boldsymbol{x}^k, \boldsymbol{d}^k; \boldsymbol{w}_i^k)}, \tag{6}$$

where $\tau$ is a positive scaling factor akin to the inverse temperature in Gibbs distribution. Recall Assumption 1 that $\boldsymbol{x}^k = \ddot{\boldsymbol{x}}^k + \boldsymbol{d}^k$. The numerator, *i.e.*, the unnormalized effect, can be implemented as logits $f(\boldsymbol{x}^k, \boldsymbol{d}^k; \boldsymbol{w}_i^k) = (\boldsymbol{w}_i^k)^\top(\ddot{\boldsymbol{x}}^k + \boldsymbol{d}^k) = (\boldsymbol{w}_i^k)^\top \boldsymbol{x}^k$, and the denominator is a normalization term (or propensity score [49]) that only balances the magnitude of the variables: $g(\boldsymbol{x}^k, \boldsymbol{d}^k; \boldsymbol{w}_i^k) = \|\boldsymbol{x}^k\| \cdot \|\boldsymbol{w}_i^k\| + \gamma\|\boldsymbol{x}^k\|$, where the first term is a class-specific energy and the second term is a class-agnostic baseline energy.

Putting the above all together, the logit calculation for $P(Y = i|do(X = \boldsymbol{x}))$ can be formulated as:

$$[Y = i|do(X = \boldsymbol{x})] = \frac{\tau}{K} \sum_{k=1}^{K} \frac{(\boldsymbol{w}_i^k)^\top(\ddot{\boldsymbol{x}}^k + \boldsymbol{d}^k)}{(\|\boldsymbol{w}_i^k\| + \gamma)\|\boldsymbol{x}^k\|} = \frac{\tau}{K} \sum_{k=1}^{K} \frac{(\boldsymbol{w}_i^k)^\top \boldsymbol{x}^k}{(\|\boldsymbol{w}_i^k\| + \gamma)\|\boldsymbol{x}^k\|}. \tag{7}$$

Interestingly, this model also explains the effectiveness of normalized classifiers like cosine classifier [50, 51]. We will further discuss it in Section 4.4.

| Methods | Two-stage | Re-balancing ($do(D)$) | De-confound ($do(X)$) | Direct Effect |
|---|---|---|---|---|
| Cosine [50, 51] | - | - | ✔ | - |
| LDAM [12] | - | ✔ | ✔ | CDE |
| OLTR [9] | ✔ | ✔ | - | NDE |
| BBN [10] | ✔ | ✔ | - | NDE |
| Decouple [11] | ✔ | ✔ | - | NDE |
| EQL [17] | - | ✔ | - | - |
| Our method | - | - | ✔ | TDE |

Table 1: Revisiting the previous state-of-the-arts in our causal graph. CDE: Controlled Direct Effect. NDE: Natural Direct Effect. TDE: Total Direct Effect.

## 4.2 Total Direct Effect Inference

After the de-confounded training, the causal graph is now ready for inference. The TDE of $X \rightarrow Y$ in Eq. (2) can thus be depicted as in Figure 3. By applying the counterfactual consistency rule [52], we have $[Y_d = i|do(X = x)] = [Y = i|do(X = x)]$. This indicates that we can use Eq. (7) to calculate the first term of Eq. (2). Thanks to Assumption 1, we can disentangle $x$ by $x = \ddot{x} + d$, where $d = \|d\| \cdot \hat{d} = cos(x, \hat{d})\|x\| \cdot \hat{d}$. Therefore, we have $[Y_d = i|do(X = x_0)]$ that replaces the $\ddot{x}$ in Eq. (7) with zero vector, just like "cheating" the model with a null input but keeping everything else unchanged. Overall, the final TDE calculation for Eq. (2) is

$$TDE(Y_i) = \frac{\tau}{K}\sum_{k=1}^{K}\left(\frac{(\boldsymbol{w}_i^k)^\top \boldsymbol{x}^k}{(\|\boldsymbol{w}_i^k\| + \gamma)\|\boldsymbol{x}^k\|} - \alpha \cdot \frac{cos(\boldsymbol{x}^k, \hat{\boldsymbol{d}}^k) \cdot (\boldsymbol{w}_i^k)^\top \hat{\boldsymbol{d}}^k}{\|\boldsymbol{w}_i^k\| + \gamma}\right), \tag{8}$$

where $\alpha$ controls the trade-off between the indirect and direct effect as shown in Figure 4.

## 4.3 Background-Exempted Inference

Some classification tasks need a special "background" class to filter out samples belonging to none of the classes of interest, *e.g.*, object detection and instance segmentation use the background class to remove non-object regions [3, 24], and recommender systems assume that the majority of the items are irrelevant to a user [53]. In such tasks, most of the training samples are background and hence the background class is a good head class, whose effect should be kept and thus exempted from the TDE calculation. To this end, we propose a *background-exempted* inference that particular uses the original inference (total effect) for background class. The inference can be formulated as:

$$\arg\max_{i \in C}\begin{cases}(1 - p_0) \cdot \frac{q_i}{1 - q_0} & i \neq 0 \\ p_0 & i = 0\end{cases}, \tag{9}$$

where $i = 0$ is the background class, $p_i = P(Y = i|do(X = x))$ is the de-confounded probability that we defined in Section 4.1, $q_i$ is the softmax activated probability of the original $TDE(Y_i)$ in Eq. (8). Note that Eq. (9) adds up to 1 from $i = 0$ to $C$.

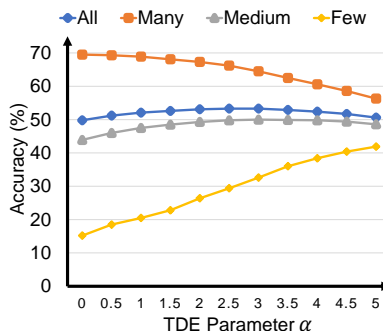

(a) Accuracy for different TDE parameter $\alpha$

Figure 4: The influence of parameter $\alpha$ in Eq. (8) on ImageNet-LT val set [9] shows how $D$ controls the head/tail preference.

## 4.4 Revisiting Two-stage Training

The proposed framework also theoretically explains the previous state-of-the-arts as shown in Table 1. Please see Appendix B for the detailed revisit for each method.

**Two-stage Re-balancing.** Naïve re-balanced training fails to retain a natural mediation $D$ that respects the inter-dependencies among classes. Therefore, the two-stage training is adopted by most of the re-balancing methods: imbalanced pre-training the backbone with natural $D$ and then balanced re-training a fair classifier with the fixed backbone for feature representation. Later, we will show that the second stage re-balancing essentially plays a counterfactual role, which reveals the reason why the stage-2 is indispensable.

| Methods | Many-shot | Medium-shot | Few-shot | Overall |
|---|---|---|---|---|
| Focal Loss[†] [28] | 64.3 | 37.1 | 8.2 | 43.7 |
| OLTR[†] [9] | 51.0 | 40.8 | 20.8 | 41.9 |
| Decouple-OLTR[†] [9, 11] | 59.9 | 45.8 | 27.6 | 48.7 |
| Decouple-Joint [11] | 65.9 | 37.5 | 7.7 | 44.4 |
| Decouple-NCM [11] | 56.6 | 45.3 | 28.1 | 47.3 |
| Decouple-cRT [11] | 61.8 | 46.2 | 27.4 | 49.6 |
| Decouple-$\tau$-norm [11] | 59.1 | 46.9 | 30.7 | 49.4 |
| Decouple-LWS [11] | 60.2 | 47.2 | 30.3 | 49.9 |
| Baseline | 66.1 | 38.4 | 8.9 | 45.0 |
| Cosine[†] [50, 51] | 67.3 | 41.3 | 14.0 | 47.6 |
| Capsule[†] [9, 54] | 67.1 | 40.0 | 11.2 | 46.5 |
| (Ours) De-confound | **67.9** | 42.7 | 14.7 | 48.6 |
| (Ours) Cosine-TDE | 61.8 | 47.1 | 30.4 | 50.5 |
| (Ours) Capsule-TDE | 62.3 | 46.9 | 30.6 | 50.6 |
| (Ours) De-confound-TDE | 62.7 | **48.8** | **31.6** | **51.8** |

Table 2: The performances on ImageNet-LT test set [9]. All models were using the ResNeXt-50 backbone. The superscript † denotes being re-implemented by our framework and hyper-parameters.

**De-confounded Training.** Technically, the proposed de-confounded training in Eq. (7) is the multi-head classifier with normalization. The normalized classifier, like cosine classifier, has already been embraced by various methods [50, 51, 9, 11] based on empirical practice. However, as we will show in Table 2, without the guidance of our causal graph, their normalizations perform worse than the proposed de-confounded model. For example, methods like decouple [11] only applies normalization in the 2nd stage balanced classifier training, and hence its feature learning is not de-confounded.

**Direct Effect.** The one-stage re-weighting/re-sampling training methods, like LDAM [12], can be interpreted as calculating Controlled Direct Effect (CDE) [23]: $CDE(Y_i) = [Y = i|do(X = \boldsymbol{x}), do(D = \boldsymbol{d}_0)] - [Y = i|do(X = \boldsymbol{x}_0), do(D = \boldsymbol{d}_0)]$, where $\boldsymbol{x}_0$ is a dummy vector and $\boldsymbol{d}_0$ is a constant vector. CDE performs a physical intervention — re-balancing — on the training data by setting the bias $D$ to a constant. Note that the second term of CDE is a constant that does not affect the classification. However, CDE removes the "bad" at the cost of hurting the "good" during representation learning, as $D$ is no longer a natural mediation generated by $X$.

The two-stage methods [10, 11] are essentially Natural Direct Effect (NDE), where the stage-2 re-balanced training is actually an intervention on $D$ that forces the direction $\hat{\boldsymbol{d}}$ do not head to any class. Therefore, when attached with the stage-1 imbalanced pre-trained features, the balanced classifier calculates the NDE: $NDE(Y_i) = [Y_{\boldsymbol{d}_0} = i|do(X = \boldsymbol{x})] - [Y_{\boldsymbol{d}_0} = i|do(X = \boldsymbol{x}_0)]$, where $\boldsymbol{x}_0$ and $\boldsymbol{d}_0$ are dummy vectors, because the stage-2 balanced classifier forces the logits to nullify any class-specific momentum direction; $do(X = \boldsymbol{x})$ as stage-1 backbone is frozen and $M \not\to X$; the second term can be omitted as it is a class-agnostic constant. Besides that their stage-1 training is still confounded, as we will show in experiments, our TDE is better than NDE because the latter completely removes the entire effect of $D$ by setting $D = \boldsymbol{d}_0$, which is however sometimes good, *e.g.*, mis-classifying "warthog" as the head-class "pig" is better than "car"; TDE admits the effect by keeping $D = \boldsymbol{d}$ as a baseline and further compares the fine-grained difference via the direct effect, *e.g.*, by admitting that "warthog" does look like "pig", TDE finds out that the tusk is the key difference between "warthog" and "pig", and that is why our method can focus on more discriminative regions in Figure 5.

## 5 Experiments

The proposed method was evaluated on three long-tailed benchmarks: Long-tailed CIFAR-10/-100, ImageNet-LT for image classification and LVIS for object detection and instance segmentation. The consistent improvements across different tasks demonstrate our broad application domain.

**Datasets and Protocols.** We followed [12, 10] to collect the long-tailed versions of CIFAR-10/-100 with controllable degrees of data imbalance ratio ($\frac{N_{max}}{N_{min}}$, where $N$ is number of samples in each category), which controls the distribution of training sets. ImageNet-LT [9] is a long-tailed subset of ImageNet dataset [4]. It consists of 1k classes over 186k images, where 116k/20k/50k for train/val/test sets, respectively. In train set, the number of images per class is ranged from 1,280 to 5, which

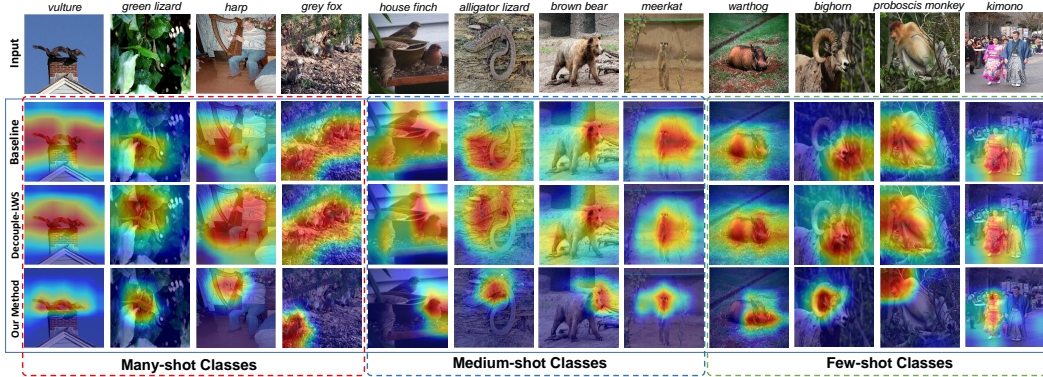

Figure 5: The visualized activation maps of the linear classifier baseline, Decouple-LWS [11] and the proposed method on ImageNet-LT using the Grad-CAM [55].

| Dataset | Long-tailed CIFAR-100 | | | Long-tailed CIFAR-10 | | |
|---|---|---|---|---|---|---|
| **Imbalance ratio** | 100 | 50 | 10 | 100 | 50 | 10 |
| Focal Loss [28] | 38.4 | 44.3 | 55.8 | 70.4 | 76.7 | 86.7 |
| Mixup [56] | 39.5 | 45.0 | 58.0 | 73.1 | 77.8 | 87.1 |
| Class-balanced Loss [13] | 39.6 | 45.2 | 58.0 | 74.6 | 79.3 | 87.1 |
| LDAM [12] | 42.0 | 46.6 | 58.7 | 77.0 | 81.0 | 88.2 |
| BBN [10] | 42.6 | 47.0 | 59.1 | 79.8 | 82.2 | 88.3 |
| (Ours) De-confound | 40.5 | 46.2 | 58.9 | 71.7 | 77.8 | 86.8 |
| (Ours) De-confound-TDE | **44.1** | **50.3** | **59.6** | **80.6** | **83.6** | **88.5** |

Table 3: Top-1 accuracy on Long-tailed CIFAR-10/-100 with different imbalance ratios. All models are using the same ResNet-32 backbone. We further adopted the same warm-up scheduler from BBN [10] for fair comparisons.

imitates the long-tailed distribution that commonly exists in the real world. The test and val sets were balanced and reported on four splits: Many-shot containing classes with $> 100$ images, Medium-shot including classes with $\geq 20 \ \& \ \leq 100$ images, Few-shot covering classes with $< 20$ images, and Overall for all classes. LVIS [7] is a large vocabulary instance segmentation dataset with 1,230/1,203 categories in V0.5/V1.0, respectively. It contains a 57k/100k train set (V0.5/V1.0) under a significant long-tailed distribution, and relatively balanced 5k/20k val set (V0.5/V1.0) and 20k test set.

**Evaluation.** For Long-tailed CIFAR-10/-100 [12, 10], we evaluated Top-1 accuracy under three different imbalance ratios: 100/50/10. For ImageNet-LT [9], the evaluation results were reported as the percentage of accuracy on four splits. For LVIS [7], the evaluation metrics are standard segmentation mask AP calculated across IoU threshold 0.5 to 0.95 for all classes. These classes can also be categorized by the frequency and independently reported as $AP_r$, $AP_c$, $AP_f$: subscripts $r, c, f$ stand for rare (appeared in $< 10$ images), common (appeared in $11 - 100$ images), and frequent (appeared in $> 100$ images). Since we can use the LVIS to detect bounding boxes, the detection results were reported as $AP_{bbox}$.

**Implementation Details.** For image classification on ImageNet-LT, we used ResNeXt-50-32x4d [2] as our backbone for all experiments. All models were trained by using SGD optimizer with momentum $\mu = 0.9$ and batch size 512. The learning rate was decayed by a cosine scheduler [57] from 0.2 to 0.0 in 90 epochs. Hyper-parameters were chosen by the performances on ImageNet-LT val set, and we set $K = 2, \tau = 16, \gamma = 1/32, \alpha = 3.0$. For Long-tailed CIFAR-10/-100, we changed the backbone to ResNet-32 and the training scheduler to warm-up scheduler like BBN [10] for fair comparisons. All parameters except for $\alpha$ are inherited from ImageNet-LT, which was set to $1.0/1.5$ for CIFAR-10/-100 respectively. For instance segmentation and object detection on LVIS, we chose Cascade Mask R-CNN framework [24] implemented by [58]. The optimizer was also SGD with momentum $\mu = 0.9$ and we used batch size 16 for a R101-FPN backbone. The models were trained in 20 epochs with learning rate starting at 0.02 and decaying by the factor of 0.1 at the 16-th and 19-th epochs. We selected the top 300 predicted boxes following [7, 17]. The hyper-parameters on LVIS were directly adopted from the ImageNet-LT, except for $\alpha = 1.5$. The main difference

| Methods | LVIS Version | AP | $AP_{50}$ | $AP_{75}$ | $AP_r$ | $AP_c$ | $AP_f$ | $AP_{bbox}$ |
|---|---|---|---|---|---|---|---|---|
| Focal Loss[28] | V0.5 | 21.1 | 32.1 | 22.6 | 3.2 | 21.1 | 28.3 | 22.6 |
| (2019 Winner) EQL [17] | V0.5 | 24.9 | 37.9 | 26.7 | 10.3 | 27.3 | 27.8 | 27.9 |
| Baseline | V0.5 | 22.6 | 33.5 | 24.4 | 2.5 | 23.0 | 30.2 | 24.3 |
| Cosine[50, 51] | V0.5 | 25.0 | 37.7 | 27.0 | 9.3 | 25.5 | 30.8 | 27.1 |
| Capsule[9, 54] | V0.5 | 25.4 | 37.8 | 27.4 | 8.5 | 26.4 | **31.0** | 27.1 |
| (Ours) De-confound | V0.5 | 25.7 | 38.5 | 27.8 | 11.4 | 26.1 | 30.9 | 27.7 |
| (Ours) Cosine-TDE | V0.5 | 28.1 | 42.6 | 30.2 | 20.8 | 28.7 | 30.3 | 30.6 |
| (Ours) Capsule-TDE | V0.5 | **28.4** | 42.1 | **30.8** | 21.1 | **29.7** | 29.6 | 30.4 |
| (Ours) De-confound-TDE | V0.5 | **28.4** | **43.0** | 30.6 | **22.1** | 29.0 | 30.3 | **31.0** |
| Baseline | V1.0 | 21.8 | 32.7 | 23.2 | 1.1 | 20.9 | 31.9 | 23.9 |
| (Ours) De-confound | V1.0 | 23.5 | 34.8 | 25.0 | 5.2 | 22.7 | **32.3** | 25.8 |
| (Ours) De-confound-TDE | V1.0 | **27.1** | **40.1** | **28.7** | **16.0** | **26.9** | 32.1 | **30.0** |

Table 4: All models are using the same Cascade Mask R-CNN framework [24] with R101-FPN backbone [59]. The reported results are evaluated on LVIS val set [7].

between image classification and object detection/instance segmentation is that the latter includes a background class $i = 0$, which is a head class used to make a binary decision between foreground and background. As we discussed in Section. 4.3, the Background-Exempted Inference should be used to retain the good background bias. The comparison between with and without Background-Exempted Inference is given in Appendix C.

**Ablation studies.** To study the effectiveness of the proposed de-confounded training and TDE inference, we tested a variety of ablation models: 1) the linear classifier baseline (no biased term); 2) the cosine classifier [50, 51]; 3) the capsule classifier [9], where $x$ is normalized by the non-linear function from [54]; 4) the proposed de-confounded model with normal softmax inference; 5) different versions of the TDE. As reported in Table (2,4), the de-confound TDE achieves the best performance under all settings. The TDE inference improves all three normalized models, because the cosine and capsule classifiers can be considered as approximations to the proposed de-confounded model. To show that the mediation effect removed by TDE indeed controls the preference towards head direction, we changed the parameter $\alpha$ as shown in Figure 4, resulting the smooth increasing/decreasing of the performances on tail/head classes, respectively.

**Comparisons with State-of-The-Art Methods.** The previous state-of-the-art results on ImageNet-LT are achieved by the two-stage re-balanced training [11] that decouples the backbone and classifier. However, as we discussed in Section 4.4, this kind of approaches are less effective or efficient. In Long-tailed CIFAR-10/-100, we outperform the previous methods [13, 12, 10] in all imbalance ratios, which proves that the proposed method can automatically adapt to different data distributions. In LVIS dataset, after a simple adaptation, we beat the champion EQL [17] of LVIS Challenge 2019 in Table 4. All reported results in Table 4 are using the same Cascade Mask R-CNN framework [24] and R101-FPN backbone [59] for fair comparison. The EQL results were copied from [17], which were trained by 16 GPUs and 32 batch size while the proposed method only used 8 GPUs and half of the batch size. We didn't compare the EQL results on the final challenge test server, because they claimed to exploit external dataset and other tricks like ensemble to win the challenge. Note that EQL is also a re-balanced method, having the same problems as [11]. We also visualized the activation maps using Grad-CAM [55] in Figure 5. The linear classifier baseline and decouple-LWS [11] usually activate the entire objects and some context regions to make a prediction. Meanwhile, the de-confound TDE only focuses on the direct effect, *i.e.*, the most discriminative regions, so it usually activates on a more compact area, which is less likely to be biased towards its similar head classes. For example, to classify a "kimono", the proposed method only focuses on the discriminative feature rather than the entire body, which is similar to some other clothes like "dress".

## 6 Conclusions

In this work, we first proposed a causal framework to pinpoint the causal effect of momentum in the long-tailed classification, which not only theoretically explains the previous methods, but also provides an elegant one-stage training solution to extract the unbiased direct effect of each instance. The detailed implementation consists of de-confounded training and total direct effect inference, which is simple, adaptive, and agnostic to the prior statistics of the class distribution. We achieved the new stage-of-the-arts of various tasks on both ImageNet-LT and LVIS benchmarks. As moving forward, we are going to 1) further validate our theory in a wider spectrum of application domains and 2) seek better feature disentanglement algorithms for more precise counterfactual effects.

## Broader Impact

The positive impacts of this work are two-fold: 1) it improves the fairness of the classifier, which prevents the potential discrimination of deep models, *e.g.*, an unfair AI could blindly cater to the majority, causing gender, racial or religious discrimination; 2) it allows the larger vocabulary datasets to be easily collected without a compulsory class-balancing pre-processing, *e.g.*, to train autonomous vehicles, by using the proposed method, we don't need collecting as many ambulance images as normal van images do. The negative impacts could also happen when the proposed long-tailed classification technique falls into the wrong hands, *e.g.*, it can be used to identify the minority groups for malicious purposes. Therefore, it's our duty to make sure that the long-tailed classification technique is used for the right purpose.

## Acknowledgments and Disclosure of Funding

This work was partially supported by the funding of NTU-Alibaba JRI and MOE AcRF Tier 2. We also want to thank Alibaba City Brain Group for the donations of GPUs, and all reviewers for their constructive comments.

## Footnotes

[1]Our code is available on `https://github.com/KaihuaTang/Long-Tailed-Recognition.pytorch`

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
