[Supplementary Material]

# Supplementary Material for "Long-Tailed Classification by Keeping the Good and Removing the Bad Momentum Causal Effect"

**Kaihua Tang[1], Jianqiang Huang[1,2], Hanwang Zhang[1]**
[1]Nanyang Technological University, [2]Damo Academy, Alibaba Group
kaihua001@e.ntu.edu.sg, jianqiang.jqh@gmail.com, hanwangzhang@ntu.edu.sg

## Abstract

This supplementary material includes: 1) additional explanations of Assumption 1; 2) revisiting previous methods in long-tailed classification; 3) the Background-Exempted Inference for object detection and instance segmentation; 4) the difference between re-balancing NDE and the proposed TDE; 5) additional ablation studies.

## A  Additional Explanations of Assumption 1

To better understand the $(M, X) \rightarrow D$ and Assumption 1, let's take a simple example. Given a learnable parameter $\theta \in \mathcal{R}^2$, and its gradients of instances for class A, B approximate to (1, 1) and (-1, 1) respectively. If each of these two classes has 50 samples, the mean gradient would be (0, 1), which is the optimal gradient direction shared by both A and B. The momentum will thus accelerate on this direction that optimizes the model to fairly discriminate two classes. However, if there are 99 samples from class A and only 1 sample from class B (long-tailed dataset), the mean gradient would be (0.98, 1). In this case, the momentum direction now approximates to the class A (head) gradients, encouraging the backbone parameters to generate head-like feature vectors, *i.e.*, creating an unfair deviation towards the head.

Since the momentum in SGD [1, 2, 3] usually dominates the gradient velocity, the effect of such a deviation is not trivial, which will eventually create the head projection $D$ on all feature vectors generated by the backbone. It's worth noting that although there are non-linear activation layers in the backbone, due to the central limit theorem [4], the overall effect of these deviated parameters is still following the normal distribution, which means we can use the moving averaged feature to approximate this head direction, *i.e.*, the Assumption 1 in the original paper.

In addition, even in a balanced dataset, the Assumption 1 still holds. Considering the above example, the mean gradient is (0, 1) for balanced A and B, which is not biased towards either direction: (1, 1) or (-1, 1). In other word, the $D$ still exists for the balanced dataset, but the $cos(\boldsymbol{x}, \hat{\boldsymbol{d}})$ should be almost the same for all classes. Therefore, the $M \rightarrow D \rightarrow Y$ won't cause any preference in the balanced dataset, which naturally allows $X \rightarrow Y$ free from the effect of $M$. It's also intuitively easy to understand, because when the dataset is balanced, the mean feature only represents the common patterns shared by all classes, *e.g.*, the $D$ in a balanced face recognition dataset is the mean face, which would be a contour of human head that not biased towards any specific face categories.

## B  Revisiting Previous Methods in Long-Tailed Classification

In this section, we will revisit the previous state-of-the-arts in two aspects: the normalized classifiers and the re-balancing strategies.

| Methods | BG-Exempted | AP | $AP_{50}$ | $AP_{75}$ | $AP_r$ | $AP_c$ | $AP_f$ | $AP_{bbox}$ |
|---|---|---|---|---|---|---|---|---|
| De-confound | ✗ | 25.7 | 38.5 | 27.8 | 11.4 | 26.1 | **30.9** | 27.7 |
| De-confound-TDE | False | 23.4 | 35.7 | 24.9 | 13.1 | 23.6 | 27.1 | 24.8 |
| De-confound-TDE | True | **28.4** | **43.0** | **30.6** | **22.1** | **29.0** | 30.3 | **31.0** |

Table 1: The results of the proposed TDE with/without Background-Exempted Inference on LVIS [13] V0.5 val set. The Cascade Mask R-CNN framework [14] with R101-FPN backbone [15] is used.

**Normalized Classifiers.** The normalized classifiers [5, 6, 7, 8] have already been widely adopted in long-tailed classification based on empirical practice. As we discussed in the Section 4, the correctly applied normalized classifiers are approximations of the proposed de-confounded training. However, without the guidance of the proposed causal framework, most of them are not utilized in a proper way. We define the general normalized classifier as the following equation:

$$\arg\max_{i\in C} P(Y = i|X = \boldsymbol{x}) = \frac{e^{\boldsymbol{z}_i}}{\sum_{c=1}^{C} e^{\boldsymbol{z}_c}}, \quad \text{where} \quad z_i = \frac{\tau}{K} \sum_{k=1}^{K} \frac{(\boldsymbol{w}_i^k)^{\top}\boldsymbol{x}^k}{N(\boldsymbol{x}^k, \boldsymbol{w}_i^k)}. \tag{1}$$

Since in most of the previous methods, $K$ is set to 1, so we slightly abuse the notation to omit the superscript $k$ for simplicity.

The cosine classifier [5, 6] is defined based on the cosine similarity, which has $N(\boldsymbol{x}, \boldsymbol{w}_i) = \|\boldsymbol{x}\| \cdot \|\boldsymbol{w}_i\|$. It is commonly used in the tasks like few-shot learning [9]. In Table 2,3 of original paper, we have proved its effectiveness in the long-tailed classification. The capsule classifier is proposed by Liu *et al.* [8] as the replacement of vanilla cosine classifier in OLTR. It changes the $l2$ norm of $\boldsymbol{x}$ into the squashing non-linear function proposed in Capsule Network [10], which allows the normalized $\boldsymbol{x}$ having a magnitude range from 0 to 1, representing the probability of $\boldsymbol{x}$ in its direction. The final normalization term can thus be defined as $N(\boldsymbol{x}, \boldsymbol{w}_i) = (\|\boldsymbol{x}\| + 1) \cdot \|\boldsymbol{w}_i\|$. However, the OLTR [8] doesn't use it to de-confound the visual feature. Instead, its $\boldsymbol{x}$ is the joint embedding of the feature vector and an attentive memory vector. The Decouple [7] also invents two different types of normalized classifiers: $\tau$-norm classifier and Learnable Weight Scaling (LWS) classifier. They empirically found that the $l2$ norm of $\boldsymbol{w}_i$ is not uniform in the long-tailed dataset, and has a positive correlation with the number of training samples for class $i$, as shown in Figure 1.

(a) Magnitude of $w_i$ for each class $i$

Figure 1: The magnitudes of classifier weights $\|\boldsymbol{w}_i\|$ for each class after training with momentum $\mu = 0.9$, where $i$ is ranking by the number of training samples in a descending order.

Therefore, their normalized classifiers only normalize the $\boldsymbol{w}_i$: the $\tau$-norm classifier is defined as $N(\boldsymbol{x}, \boldsymbol{w}_i) = \|\boldsymbol{w}_i\|^{\tau}, \tau \in [0, 1]$ while LWS is $N(\boldsymbol{x}, \boldsymbol{w}_i) = g_i$, where $g_i$ is a learnable parameter. Yet, these decouple classifiers fail to de-confound the $M \to X$ for two reasons: 1) they don't considering the confounding effect on $\boldsymbol{x}$; 2) they only apply the normalized classifiers on the 2nd stage when the backbone has already been frozen.

**Re-balancing Strategies.** Both OLTR [8] and Decouple [7] adopt the same class-aware sampler in their 2nd stage training, which forces each class to contribute the same number of samples regardless of the size. To dynamically combine the two training stages, the BBN [11] utilizes a bilateral-branch design to smoothly transfer the sampling strategy from the imbalanced branch to the re-balancing branch, where two branches share the same set of parameters but learn from different sampling strategies, which has the same spirit as two-stage design in OLTR [8] and Decouple [7]. As to the EQL [12], since the re-sampling is complicated in the object detection and instance segmentation tasks, where objects from different classes co-exist in one image, they choose the re-weighted loss to balance the contributions of different classes.

Figure 2: A simple one-dimensional binary classification example of conventional classifier, one-/two-stage re-balancing classifiers, and the proposed TDE.

## C  Background-Exempted Inference

The results with and without Background-Exempted Inference are reported in Table 1. As we can see, the Background-Exempted strategy successfully prevents the TDE from hurting the foreground-background selection. It is the key to apply TDE in tasks like object detection and instance segmentation that include one or more legitimately biased head categories, *i.e.*, this strategy allows us to conduct TDE on a selected subset of categories.

## D  The Difference Between Re-balancing NDE and The Proposed TDE

In this section, we will further discuss the relationship between two-stage re-balancing NDE and the proposed TDE. As we discussed in Section 4.3 of original paper, the 2nd-stage re-balanced classifier essentially calculates the $NDE(Y_i) = [Y_{d'} = i | do(X = x)] - [Y_{d'} = i | do(X = x')]$, where the second term can be omitted because $x'$ is a dummy vector and the moving averaged $d'$ in a balanced set won't point to any specific classes, so it is actually a constant offset. Therefore, the crux of understanding the NDE would be why the 2nd-stage re-balanced training equals to the first term $[Y_{d'} = i | do(X = x)]$. It is because when the backbone is frozen, it breaks the dependency between $M \rightarrow X$, which is a straightforward implementation of causal intervention $do(X = x)$. The original OLTR [8] violates this intervention by fine-tuning the backbone parameters in the 2nd stage, and it thus performs much worse than the Decouple-OLTR in the Table 2 of original paper, which freezes the backbone parameters. Meanwhile, the balanced re-sampling also brings a fair $d'$ as we discussed in the third paragraph of Section A.

To better illustrate both the similarity and the difference between re-balancing NDE and the proposed TDE, we constructed a one-dimensional binary classification example for conventional classifier, one-/two-stage re-balancing classifiers, and the proposed TDE in Figure 2, where the gaussian distribution curve represents the feature distribution generated by the backbone, and the 0 point is the classifier's decision boundary. The conventional classifier and one-stage re-balancing are fundamentally problematic, because they either cause the mismatching in the inference or learn a bad backbone model. In the meantime, both two-stage re-balancing and the proposed TDE are able to correctly remove the bias by proper adjustments. The 2nd-stage re-balanced training (NDE) fixes the backbone parameters $do(X = x)$ learnt from 1st-stage imbalanced training, *i.e.*, the frozen curve in the image, and then re-samples an artificially balanced data distribution to create a fair $d'$. The overall re-balancing NDE can be considered as subtracting a bias offset from original decision boundary. Meanwhile, the proposed TDE removes the bias effect (head projection) from feature vectors. Both two types of adjustments can properly remove the head bias in this example. That's why TDE and NDE should be theoretically identical in the long-tailed classification scenario. However, the 2nd-stage re-balancing NDE has two disadvantages: 1) its adjustment requires an additional training stage to fine-tune the classifier weights, which relies on the accessibility of data distribution; 2) if non-linear modules are applied to the feature vectors, *e.g.*, a global context layer that conducts

| $K$ | $\tau$ | $\gamma$ | $\alpha$ | Many-shot | Medium-shot | Few-shot | Overall |
|---|---|---|---|---|---|---|---|
| **1** | 16.0 | 1/32.0 | ✗ | 69.8 | 42.8 | 14.9 | 49.4 |
| **4** | 16.0 | 1/32.0 | ✗ | 69.0 | 42.3 | 13.1 | 48.6 |
| 2 | **8.0** | 1/32.0 | ✗ | 69.5 | 31.3 | 1.6 | 42.0 |
| 2 | **32.0** | 1/32.0 | ✗ | 68.6 | 41.3 | 13.0 | 47.9 |
| 2 | 16.0 | **1/16.0** | ✗ | 69.3 | **44.0** | 14.2 | 49.7 |
| 2 | 16.0 | **1/64.0** | ✗ | **69.9** | 43.3 | 14.7 | 49.6 |
| **2** | **16.0** | **1/32.0** | ✗ | 69.5 | 43.9 | **15.2** | **49.8** |
| 2 | 16.0 | 1/32.0 | **2.5** | **66.2** | 49.8 | 29.4 | **53.3** |
| 2 | 16.0 | 1/32.0 | **3.0** | 64.5 | **50.0** | 32.6 | **53.3** |
| 2 | 16.0 | 1/32.0 | **3.5** | 62.5 | 49.9 | **36.0** | 52.9 |

Table 2: Hyper-parameters selection based on performances of ImageNet-LT val set, where ✗ for $\alpha$ means that TDE inference is not included. The backbone we used here is ResNeXt-50-32x4d.

| Methods | #heads $K$ | Many-shot | Medium-shot | Few-shot | Overall |
|---|---|---|---|---|---|
| Cosine[†] [5, 6] | 1 | 67.3 | 41.3 | 14.0 | 47.6 |
| Cosine[†] [5, 6] | 2 | 67.5 | 42.1 | 14.1 | 48.1 |
| Capsule[†] [8, 10] | 1 | 67.1 | 40.0 | 11.2 | 46.5 |
| Capsule[†] [8, 10] | 2 | 67.7 | 41.3 | 12.6 | 47.6 |
| (Ours) De-confound | 1 | 67.3 | 41.8 | 15.0 | 47.9 |
| (Ours) De-confound | 2 | **67.9** | 42.7 | 14.7 | 48.6 |
| (Ours) Cosine-TDE | 1 | 61.8 | 47.1 | 30.4 | 50.5 |
| (Ours) Cosine-TDE | 2 | 63.0 | 47.3 | 31.0 | 51.1 |
| (Ours) Capsule-TDE | 1 | 62.3 | 46.9 | 30.6 | 50.6 |
| (Ours) Capsule-TDE | 2 | 62.4 | 47.9 | 31.5 | 51.2 |
| (Ours) De-confound-TDE | 1 | 62.5 | 47.8 | **32.8** | 51.4 |
| (Ours) De-confound-TDE | 2 | 62.7 | **48.8** | 31.6 | **51.8** |

Table 3: The performances of cosine classifier [5, 6] and capsule classifier [8, 10] under different number of head $K$ on ImageNet-LT test set. Other hyper-parameters are fixed.

interactions among all objects $\{\boldsymbol{x}_j\}$ in an image, the NDE can only remove a linear approximation of this non-linear activated head bias, while the TDE would be able to maintain the natural interactions of features in both original logit term and the subtracted counterfactual term. It explains why the Decouple-OLTR in Table 2 of original paper doesn't perform as good as Decouple-$\tau$-norm or Decouple-LWS, because OLTR involves non-linear interactions between feature vectors and memory vectors, so a linear adjustment on classifier's decision boundary cannot completely remove the head bias.

# E    Additional Ablation Studies

The hyper-parameters used in original paper are selected according to the performances on ImageNet-LT val set as shown in Table 2. To further study the multi-head strategy on different normalized classifiers, we tested the $K = 2$ on cosine classifier [5, 6] and capsule classifier [8, 10] in Table 3. It

| Methods | Backbone | Many-shot | Medium-shot | Few-shot | Overall |
|---|---|---|---|---|---|
| Baseline | ResNeXt-50 | 66.1 | 38.4 | 8.9 | 45.0 |
| De-confound | ResNeXt-50 | 67.9 | 42.7 | 14.7 | 48.6 |
| De-confound-TDE | ResNeXt-50 | 62.7 | 48.8 | 31.6 | 51.8 |
| Baseline | ResNeXt-101 | 68.7 | 42.5 | 11.8 | 48.4 |
| De-confound | ResNeXt-101 | **68.9** | 44.3 | 16.5 | 50.0 |
| De-confound-TDE | ResNeXt-101 | 64.7 | **50.0** | **33.0** | **53.3** |

Table 4: The performances of the proposed method under different backbones in ImageNet-LT test set.

| Methods | Backbone | AP | $AP_{50}$ | $AP_{75}$ | $AP_r$ | $AP_c$ | $AP_f$ | $AP_{bbox}$ |
|---|---|---|---|---|---|---|---|---|
| Baseline | R101-FPN | 22.6 | 33.5 | 24.4 | 2.5 | 23.0 | 30.2 | 24.3 |
| De-confound | R101-FPN | 25.7 | 38.5 | 27.8 | 11.4 | 26.1 | 30.9 | 27.7 |
| De-confound-TDE | R101-FPN | 28.4 | 43.0 | 30.6 | **22.1** | 29.0 | 30.3 | 31.0 |
| Baseline | X101-FPN | 26.4 | 39.5 | 28.4 | 7.4 | 28.1 | 32.0 | 28.5 |
| De-confound | X101-FPN | 28.4 | 41.9 | 30.6 | 13.3 | 29.5 | **32.9** | 30.5 |
| De-confound-TDE | X101-FPN | **30.4** | **45.1** | **32.9** | 21.1 | **31.8** | 32.3 | **33.1** |

Table 5: The performances of the proposed method under different backbones in LVIS V0.5 val set.

| Methods | AP | $AP_{50}$ | $AP_{75}$ | $AP_r$ | $AP_c$ | $AP_f$ |
|---|---|---|---|---|---|---|
| Baseline | 19.4 | 29.8 | 20.6 | 3.9 | 21.9 | 30.8 |
| De-confound | 20.8 | 31.8 | 22.1 | 7.4 | 22.7 | **31.2** |
| De-confound-TDE | **23.0** | **35.2** | **24.1** | **12.7** | **24.5** | 30.7 |

Table 6: The single model performances of the proposed method on LVIS V0.5 evaluation test server [16].

proves that the advantage of the proposed de-confounded model doesn't come from larger K, and the multi-head fine-grained sampling can generally improves the de-confounded training, no matter what kind of normalization function we choose.

As shown in Table 4,5, we tested the proposed method on different backbones. After equipped with ResNeXt-101-32x4d and ResNeXt-101-64x4d [17] for ImageNet-LT [8] and LVIS [13] V0.5, respectively, the proposed method gains additional improvements. In ImageNet-LT dataset, we changed some hyper-parameters ($K = 4, \gamma = 1/64.0$) and increased the training epochs to 120, because of the significantly increased number of model parameters. The hyper-parameters for LVIS are still the same as original paper.

We also reported the performances of the proposed method on LVIS V0.5 evaluation test server [16] in Table 6, where we used ResNeXt-101-64x4d backbone and the original hyper-parameters. It's worth noting that these are single model performances, which neither exploited external dataset nor utilized any model enhancement tricks.