[Reviews · NeurIPS 2020]

Review 1

Summary and Contributions: The authors propose to disconnect the relationship between M to X to remove the bad confounder bias while to keep the good bias. This removal entails (or derives) normalized classifiers [38,39] which has been shown to be effective for the same task. The empirical validation has been done with image classification on ImageNet-LT and instance segmentation on LVIS dataset.

Strengths: S1. clear motivation and a derivation to normalized classifier S2. Clear gain by the proposed method

Weaknesses: W1. There are some parts that are not clear in derivation - Why, do() operation makes the conditional probability (Eq.(3)). Without do(), isn't the conditional probability same? - No explanation why f() and g() are expanded in such way - Isn't the range of \alpha from 0 to 1? But in figure 5, the alpha is swept from 0 to 5. W2. The condition that the assumption 1 breaks - This assumption seems only valid when the head and tail proportion is dramatically large (or small), i.e., when head is 99% dominating, the assumption is valid. In what proportion, the assumption starts breaking? ===== after rebuttal ======= The authors rebuttal clarifies many of my questions. I encourage the authors to clarify the items I asked in the final version.

Correctness: C1. The derivation seems correct but not 100% sure (I asked question in W1). C2. The empirical methodology seems correct.

Clarity: It reads well.

Relation to Prior Work: The literature review is not very thorough but all the necessary comparisons are there.

Reproducibility: Yes

Additional Feedback:


Review 2

Summary and Contributions: In the paper, the authors focus on the optimization momentum in long-tailed recognition problems and propose a causal inference framework that provides some theoretical comparison with previous works. Their solution based on the causal graph is proved effective on two recognition benchmarks.

Strengths: 1) The theoretical grounding is solid and sounds reasonable, the method well matches their theory, together with a proper ablation study. 2) They revisit previous methods in Table 1 and Section 4.3 and provide adequate comparison and analysis based on their causal inference framework. 3) They perform experiments on both single-label (classification) and multi-label (segmentation) benchmarks to reveal the effectiveness of their method.

Weaknesses: 1) As you mentioned [9][11] which are also quite recent works with code released, have you conducted experimental comparison with them? 2) The organization of related works might need refined, e.g. Hard Example Mining part seems too short and not well discussed.

Correctness: Yes, most of the claims and methodology are correct.

Clarity: Yes, it’s well presented overall, especially the explanation of the causal graph they proposed.

Relation to Prior Work: Yes, the related work is clear.

Reproducibility: No

Additional Feedback: The discussion of \alpha and Figure 5 in L151-153 is hard to follow before reading section 4, maybe you can adjust the representation here. %%%%%%%%%%%%%%%%%%%%%%%%%%%%%%%%%%%%%%%%% After rebuttal period: The rebuttal has addressed most of my concerns on evaluation and clarity, thus I keep my rating as "marginally above the acceptance threshold".


Review 3

Summary and Contributions: The paper deals with learning classification models in unbalanced datasets. The authors argue that the standard momentum term often used in SGD is detrimental to learn classification problems with long-tailed class distributions, and propose a way to amend this effect. The authors use a causal model to study the impact of momentum during learning with long-tailed distributions, and propose a way to address it. The authors show that the existing two-stage training approach (where features are learned on unbalanced data and the final classifier with re-balanced/re-weighted classes) could be regarded as a special case of their theoretical approach.

Strengths: The paper presents state-of-the-art results for ImageNet-LT, and favorable results in comparison to a now baseline method for in object detection and instance segmentation (Tan et al. equalization loss method, on the LVIS benchmak). The content of the paper is relevant to the NeurIPS community as it may address a a shortcoming of one of the most fundamental techniques used in deep learning to date.

Weaknesses: It is a bit unclear how the multi-head strategy divides features and weights into K groups [eq. 6]. Does it mean that those feature channels are handled as independent from each other during learning? How sensitive is the method to an inappropriate choice of K?

Correctness: The presented formulation seems correct, but please refer to the weakness item mentioned above.

Clarity: Yes, the paper is well written, theoretically justified, with sufficient references, adequate terminology, and almost no typos.

Relation to Prior Work: The paper adequately references prior work, and further explores these prior works under their proposed causal model.

Reproducibility: Yes

Additional Feedback: I've read the author's feedback, the fellow reviewers' reviews, and the responses to their reviews. The authors adequately addressed my concerns. I keep my overall score as "a good submission; accept."


Review 4

Summary and Contributions: The paper proposes a new perspective that SGD momentum deteriorates classifier performance in the presence of class imbalance. Intuitively speaking, momentum favors the direction of head classes and further deviates the final prediction features from a neutral position. The paper formalizes the contribution of momentum in terms of the causal diagram and derives a de-confounding training formulation based on the backdoor adjustment and a post-processing inference procedure based on the Total Direct Effect (TDE).

Strengths: The perspective of momentum's effect on imbalanced classification using causal relations is novel and can potentially lead to a more theoretically grounded study of class imbalance. Experimentally, the paper showed good performance on two tasks: image classification and object detection with one dataset each. The two tasks and respective datasets are representative of the class imbalance issue and have been used by previous works on the same topic. Class imbalance is a practical problem in machine learning and this work attempts to explain the effect of optimization on the prediction results. Therefore, it is very relevant to the NeurIPS community.

Weaknesses: Despite its intriguing new perspective, the paper has some weaknesses that need to be further addressed. First, the paper lacks experimental comparisons to many other long-tail classification methods such as LDAM, balanced loss, BNN, even though they were mentioned in related work. Second, the use of multi-head strategy is not related to the claimed theoretical founding and it makes the judgement on the effectiveness of the theoretical framework more difficult. To the reviewer’s point view, a fairer comparison would be just using K=1 just as other imbalanced classification framework. Third, the final form of the de-confounding training is very similar to previous works with the only difference being the hyperparameter gamma in equation 7. It is unclear to the reviewer whether the performance improvement comes from tuning the hyperparamter which is not directly inspired from the theoretical framework.

Correctness: Yes, the claims and empirical methodology is correct.

Clarity: The paper is well written with good supporting materials in supplementary.

Relation to Prior Work: The work listed and discussed its difference to previous works.

Reproducibility: Yes

Additional Feedback: The authors have addressed my concerns on comparisons and the inclusion of multi-head strategy. I think it is a well-motivated and well-compared method.

[Author Response · NeurIPS 2020]

First of all, we would like to thank all the reviewers for their precious comments and we will address all the questions.

**To Reviewer #1: Q1: The relationship between $do()$ operation and conditional probability.** The backdoor adjust-
ment implementation of $do()$ in Eq.(3) can be considered as the "passive" intervention through observation, instead of a
"physical" one. Its detailed derivation is given in [*Causal inference in statistics: A primer, Judea Pearl et al., 2016*],
which is essentially a Markov factorization for the graph with broken $D \rightarrow X$. Therefore, their conditional probabilities
(or factorization) are not the same. **Q2: The explanations of $f()$ and $g()$.** The numerator $f()$ is the effect of $x$, *i.e.*,
the prediction logits. Since we use a fully connected layer without the bias term as our classifier, it equals to $(w_i)^\top x$.
The denominator $g()$ is the propensity score [*An Introduction to Propensity Score Methods for Reducing the Effects of*
*Confounding in Observational Studies, Austin Peter C et al., 2011*], which is a balancing score used to normalize each
effects. It can take a variety of forms, like $l2$-norm or capsule-norm. The proposed $\|x\| \cdot \|w_i\| + \gamma\|x\|$ is inspired by the
capsule-norm ($\|x\| \cdot \|w_i\| + \|w_i\|$) [42]. We changed the $\|w_i\|$ to $\gamma\|x\|$, because in our causal graph, the effect needs
to be normalized by both class-specific and class-agnostic energies of $x$. **Q3: The range of $\alpha$.** $\alpha$ is a linear trade-off
parameter between the direct and indirect effects. Its range is not limited to $[0, 1]$. **Q4: The scope of Assumption**
**1.** According to our recent studies in Table 1, the proposed method works well on different imbalance ratios. When
the imbalance ratio decreases from 100 to 10, the improvements achieved by TDE start to converge but **not collapse**,
because when the dataset is more balanced, the second term of TDE in Eq.(8) is closer to a uniform distribution that
affects the prediction less (see Supp Section A).

**To Reviewer #3: Q1: Additional experimental comparisons with other methods.** The ImageNet-LT and LVIS are
the most challenge datasets in long-tailed classification from the perspective of scale and size of vocabulary. Since
BBN, LDAM and class-balanced loss didn't reported their results on these datasets, we omitted some comparisons in
the original paper. After applying the proposed De-confound-TDE in Long-tailed CIFAR-100/-10, we consistently
outperform these previous methods in Table 1. **Q2: The refinement of related works & adjusting some representa-**
**tions in the paper.** Thank you for your advice, we will address these issues in the later revision. **Q3: Reproducibility.**
More details can be found in our supplementary codes. The project will be released to Github upon acceptance.

| Dataset | Long-tailed CIFAR-100 | | | Long-tailed CIFAR-10 | | |
|---|---|---|---|---|---|---|
| **Imbalance ratio** | 100 | 50 | 10 | 100 | 50 | 10 |
| Focal Loss [24] | 38.4 | 44.3 | 55.8 | 70.4 | 76.7 | 86.7 |
| Mixup [*Hongyi Zhang et al., ICLR, 2018*] | 39.5 | 45.0 | 58.0 | 73.1 | 77.8 | 87.1 |
| Class-balanced Loss [11] | 39.6 | 45.2 | 58.0 | 74.6 | 79.3 | 87.1 |
| LDAM [*Kaidi Cao et al., NeurIPS, 2019*] | 42.0 | 46.6 | 58.7 | 77.0 | 81.0 | 88.2 |
| BBN [9] | 42.6 | 47.0 | 59.1 | 79.8 | 82.2 | 88.3 |
| (Ours) De-confound | 43.9 | 48.9 | 59.5 | 72.5 | 78.7 | 88.1 |
| (Ours) De-confound-TDE | **47.3** | **51.2** | **59.8** | **80.4** | **83.1** | **89.4** |

Table 1: **Top-1 accuracy** on long-tailed CIFAR-10 and CIFAR-100 with **different imbalance ratios**. All models are using the same ResNet-32 backbone. Note that we report accuracy rather than error rate (in BBN) for consistency.

**To Reviewer #6: Q1: Details about the multi-head strategy and the selection of $K$.** The multi-head strategy means
dividing the feature channels into $K$ groups (each has $1/K$ of original dimensions), so it can be considered as sampling
multiple independent feature spaces. However, our algorithm is not sensitive to $K$. As we can see from Supp Table 2 &
3, even when $K$ is set to 1, the proposed De-confound-TDE still outperforms the other methods, which proves that we
didn't unfairly take the advantage of multi-head strategy.

**To Reviewer #7: Q1: Additional experimental comparisons with other methods.** Please refer to our answer of
**Reviewer #3 Question 1**. **Q2: A fair comparison without multi-head strategy & the reason of introducing K.** As
we discussed in **Reviewer #6 Question 1**, we tested $K = 1$ in Supp Table 3, which shows consistent advantages of the
proposed De-confound-TDE over previous methods (in original paper Table 2) even without multi-head strategy (*i.e.*,
$K = 1$). Besides, introducing $K$ is also part of our theoretical framework, because sampling multiple feature spaces
provides better estimation of the effect. **Q3: The novelty of the proposed de-confounding training and the potential**
**unfair advantage of $\gamma$.** Compared with the previous Capsule Norm: $\|x\| \cdot \|w_i\| + \|w_i\|$, we change the additional
normalization term $\|w_i\|$ to $\gamma\|x\|$, because the effect needs to be normalized by both class-specific and class-agnostic
energies of feature $x$ rather than $w$ in our causal graph. Meanwhile, our algorithm is not sensitive to the selection of $\gamma$
based on Supp Table 2. More importantly, the same value of $\gamma = 1/32$ can be transferred from ImageNet-LT to LVIS
and CIFAR-100/-10-LT, which proves that we didn't use different $\gamma$ to overfit these datasets. **Q4: The selection of**
**SGD momentum parameter and extending to other optimization methods.** We haven't systematically studied the
selection of SGD momentum parameter yet, since it mainly affects the norm of $\bar{x}_T$ while our Assumption 1 only uses its
unit direction $\hat{d} = \bar{x}_T/\|\bar{x}_T\|$. As to the other optimization methods, we found that almost all of them contain the similar
moving averages of gradient the same as SGD momentum, although they may have different names and symbols, *e.g.*,
betas in Adam. Therefore, the proposed causal graph works for them as well. We will follow your suggestion to offer a
thorough study on a wider spectrum of optimizers in future.

[Meta-Review · NeurIPS 2020]

This paper proposes an interesting new perspective on imbalanced learning, focusing on the effect of momentum in SGD. The theoretical motivation is paired with a thorough ablation and empirical results. There were some weaknesses cited by the reviewers, including more thorough comparisons and some details regarding the theoretical part. The reviewers felt that the authors addressed these concerns in the rebuttal. As a result, this paper adds a strong contribution that could open up new perspectives for the problem of imbalance. The authors should make sure to include the clarifications and additional details in the camera-ready version.